# Awareness, Attitudes, and Perceptions of Oral Healthcare among First Year Dental, Medical, and Nursing Students

**DOI:** 10.3390/dj11070169

**Published:** 2023-07-12

**Authors:** Tassya Lay, Fadiza Nurchasanah, Dessie Wanda, Indriasti Indah Wardhany, Rulliana Agustin, Satoru Haresaku, Yuniardini Septorini Wimardhani, Masita Mandasari

**Affiliations:** 1Dentistry Study Program, Faculty of Dentistry, Universitas Indonesia, Jakarta 10430, Indonesia; tassya.lay@ui.ac.id (T.L.); fadizaaa@gmail.com (F.N.); 2Department of Pediatric Nursing, Faculty of Nursing, Universitas Indonesia, Jakarta 16424, Indonesia; dessie@ui.ac.id; 3Department of Oral Medicine, Faculty of Dentistry, Universitas Indonesia, Jakarta 10430, Indonesia; indriasti.indah@ui.ac.id (I.I.W.); juneardini@gmail.com (Y.S.W.); 4Department of Medical Education, Faculty of Medicine, Universitas Indonesia, Jakarta 10430, Indonesia; rulliana.agustin@gmail.com; 5Department of Nursing, Fukuoka Nursing College, Fukuoka 814-0193, Japan; haresaku@college.fdcnet.ac.jp

**Keywords:** attitudes, awareness, interprofessional education, oral healthcare, perceptions

## Abstract

Background: It has been reported that poor oral health can worsen general health conditions. Good collaboration between health professionals is important to provide proper oral healthcare. Thus, there is a need for oral healthcare interprofessional education (IPE). This study aimed to determine the baseline level of awareness, attitudes, and perceptions of oral healthcare among first-year students of dentistry, medicine, and nursing at Universitas Indonesia. Method and Participants: A cross-sectional descriptive analytical study using a previously published questionnaire was conducted involving 442 students, consisting of dental students (DS), medical students (MS), and nursing students (NS) in Universitas Indonesia. Results: Most students have shown good oral healthcare awareness, attitudes, and perception with no statistically significant difference between the groups (*p* < 0.05). The majority of the students did not perceive that (1) geriatric and nursing knowledge are required to practice oral care, (2) oral healthcare should be provided in cancer hospitals, and (3) oral healthcare can prevent cardiovascular disease and improve anorexia. Conclusions: This study showed that there were aspects of oral healthcare that should be improved in all student groups. Thus, oral healthcare IPE should be given to all students working in healthcare to develop knowledge and interprofessional collaboration in oral healthcare.

## 1. Introduction

Oral health refers to the health of teeth, gums, and the entire mouth–face system that allows us to smile, talk, and chew; thus, good oral health can improve the nutritional status of patients [1,2]. The World Oral Health Report stated that there was a relationship between oral health and general health, as poor oral health can worsen general health conditions [3]. Especially in an aging society, there will be more people with medical conditions and often hospitalized. They tend to have oral health problems such as lower oral hygiene and oral function as well as oral diseases such as caries and periodontal disease and are at risk for ventilator-associated pneumonia (VAP) [1,4,5,6,7]. Moreover, critical patients may have very specific care needs and demand the highest standards of professional care [8]. It has been shown that patients who received proper oral care showed acceptable oral health status [9].

Good oral healthcare necessitates cooperation amongst healthcare professionals since each professional can help in assessing patients’ oral conditions to prevent more serious complications as well as initiate interprofessional consultations for appropriate treatment [4,10,11]. Oral health education is not just important for dental students and professionals but also for other health students and professionals who will directly interact with patients, such as nurses or doctors, to promote collaborative oral healthcare [12,13,14]. It is suggested that good collaborative oral healthcare comes from interprofessional education since it provides training and teamwork experience especially given early and regularly during their education [14,15,16].

There have been reports of the low level of oral healthcare knowledge in medical and nursing students in India, Jordan, Japan, and Nigeria which is attributed to the lack of oral health education and training in their curriculum [17,18,19,20]. Our university has interprofessional education (IPE) of health collaboration for medical, dental, nursing, pharmacy, and public health students, which was organized in the second semester of the undergraduate program and in the final semester of their professional (clinical education) program. However, it was still lacking in regard to oral healthcare. Therefore, this study aimed to explore the level of awareness, attitudes, and perceptions about oral healthcare among students of dentistry, medicine, and nursing at Universitas Indonesia to provide the background information for future improvement of oral healthcare content in IPE.

## 2. Respondents and Methods

### 2.1. Study Design and Study Sample

This study used a cross-sectional descriptive analytical research design recruiting participants of students from dentistry, medical, and nursing schools who will mostly interact directly with patients. A Google Form-based questionnaire was distributed to first-year dental students (DS), medical students (MS), and nursing students (NS) (*n* = 461) in their first semester. All students were included in this study (total sampling). Based on Slovin’s formula, minimum sample size was 214 respondents. Exclusion criterion was inability to understand Indonesian language. This study received ethical clearance from the Dental Research Ethics Committee (KEPKG) of the Faculty of Dentistry Universitas Indonesia (No. 010830921).

### 2.2. Questionnaire

The questionnaire was adapted from the questionnaire by Haresaku et al. [18] and went through cross-cultural adaptation into Indonesian language. In short, the adaptation process began after obtaining the permission of the original creator of the questionnaire in English. Two qualified English–Indonesian translators, each with medical and non-medical backgrounds, translated the questionnaire. Synthesized translations were backtranslated into English by two translators, different from the initial translators. Finally, the contents were reviewed by a team of dental, medical, and nursing academicians for finalization.

As many as 40 respondents consisting of dental, medical, and nursing students were recruited for reliability and validity tests initially. The questionnaire had four sections: characteristics of the subjects (age, sex, study program), awareness toward oral healthcare (5 items), perception of oral healthcare (4 items), and attitudes regarding learning oral healthcare in lecture or practice (2 items). Response choices on items related to awareness toward oral healthcare were “very much”, “somewhat”, “a little”, “not very much”, and “not at all”.

### 2.3. Statistical Analysis

The collected data were analyzed using IBM SPSS Statistics software program (Version 25.0; IBM Corporation; Armonk, NY, USA). Collected data were analyzed according to the study of Haresaku et al. [18]. Chi-square tests were performed to show significant differences between categories.

## 3. Results

Before the main data collection, a face validity test and test–retest reliability were performed on the initial sample respondents. The reliability test score of 0.987 indicated good reliability, and all respondents reported that they did not have any difficulty understanding the questionnaire items.

At the end of the data collection, a total of 442 respondents (95.8% response rate) consisting of 117 DS, 214 MS, and 111 NS participated in this study. The response rates were 98.3%, 93%, and 99.1%, respectively. Table 1 shows the characteristics of the respondents. The majority (76.7%) of the respondents were female. Regarding age, the majority (83.7%) were older than 18 years. The distribution of respondents based on the study program was 26.5% DS, 48.4% MS, and 25.1% NS.

Table 2 shows the comparison of the NS, MS, and DS awareness of oral healthcare. Most of the students felt that they somewhat understood oral healthcare (44.8%), felt that the effectiveness of oral healthcare was generally unknown to society (49.1%), were somewhat interested in oral healthcare (39.1%), stated that they would somewhat practice oral healthcare after obtaining professional qualifications (40.3%), and around 50–74% of their duties would account for oral healthcare (31.2%). There were significant differences between groups regarding the interest in oral healthcare, the interest in practicing oral healthcare after obtaining professional qualifications, and the proportion of duties that would be used to practice oral healthcare after obtaining professional qualifications had the highest proportions in DS groups (*p* < 0.05).

Perceptions of oral healthcare are shown in Table 3. Most of the students agreed that they need knowledge of general dentistry (96.8%) and general medicine (59.5%) to practice oral healthcare. More than half of the students felt that oral healthcare should be given to older adults who need nursing care (76.5%), healthy older adults (66.5%), and patients in hospital wards (57.2%). Oral healthcare was also mostly thought to be provided in long-term care facilities (71.5%), at home (61.5%), and in pediatric wards (51.6%). Less than half of the students felt that oral healthcare should be provided in recovery phase rehabilitation wards (31.7%), cancer hospitals (19.9%), hospices (48.2%), acute care hospitals (20.4%), maternity wards (10.4%), and psychiatric wards (12.7%). Most of the students knew that oral healthcare was effective in preventing dental caries (94.8%), periodontal disease (76.2%), and general disease (59.3%), while only a minority knew that oral healthcare was effective in the prevention of cardiovascular disease (20.4%), aspiration pneumonia (14.3%), the prevention of becoming frail (21.3%), and the improvement of anorexia (21%). There were multiple significant differences between student groups in regard to the required knowledge to practice oral healthcare, individuals who should receive oral healthcare, places that should provide oral healthcare, and oral healthcare influence (*p* < 0.05). DS and MS mostly have favorable responses.

Students’ attitudes toward learning oral healthcare in lectures or practice are displayed in Table 4. The majority of students thought lectures regarding tooth brushing (75.6%), use of an interspace brush (66.3%), swabbing oral soft tissue (63.3%), support of tooth brushing (51.1%), gargling (62.4%), cleaning dentures (56.3%), removing tongue coating (75.8%), and oral management in the perioperative ward (54.1%) were necessary. In regard to learning by practice, most of the majority responses were the same with the addition of salivary gland massage (50.5%). Less than half of the students thought that oral management in the perioperative ward was necessary to learn by practice. Significant differences in attitudes were found between groups learning oral healthcare by lecture and practice (*p* < 0.05), in which the highest attitude was mostly found in the DS group.

## 4. Discussion

Most of the respondents in this study were MS since they were the largest student body among the three student groups. The students were just starting their first year when this study was conducted and had not received any oral health-related education and training. This was intentionally aimed to uncover the baseline of oral healthcare aspects in newly enrolled students. In the first semester, the classes were mainly related to personal development, bioethics and health communication, and biomedical science, which were university-organized classes.

Although most of the students were interested to learn more about oral healthcare, the interest was found to be the lowest among MS respondents. This result was in agreement with the studies conducted in India and Saudi Arabia [19,21]. Al-Hatlani et al. also found that the interests of MS (69%) and NS (76%) respondents in oral healthcare were lower than those of DS respondents (84.4%) [21]. The differences in the interest in oral healthcare between DS, MS, and NS were expected due to the differing focus of their professions and their different curricula.

More than half of the respondents felt that they do not need knowledge of geriatrics and nursing to practice oral healthcare. However, the majority of NS respondents felt that they need more knowledge of geriatrics than general medicine to practice dental and oral healthcare. This was related to their answer regarding individuals who should receive oral healthcare, of which more than half of NS respondents stated that the elderly need oral healthcare. Older adults are indeed one of the populations who need the help of nurses to practice oral healthcare the most [18]. The role of health professionals in promoting oral health in older adults is important and has been emphasized in recent European recommendations [22].

Only a small proportion of respondents knew that maintaining oral health is effective in preventing cardiovascular disease, aspiration pneumonia, care prevention, and improvement of anorexia. This is less than half of the respondents in Japan who already knew and thought to obtain such information from the literature, media, and their families [18]. This result can be used as evidence to suggest how to develop and conduct oral healthcare IPE in the university.

Merely a few respondents felt the need to learn to perform domiciliary dental care, salivary gland massage, indirect swallowing training, direct swallowing training, and language training. Swallowing training can prevent aspiration pneumonia and is included as an aspect of oral healthcare [23]. Since the respondents were first-year students, there was a possibility that they may have not known about these terminologies. Nevertheless, we would like to emphasize that these oral healthcare procedures should be taught in the curriculum.

In Universitas Indonesia, first-year NS, MS, and DS have IPE on the basics of biomedical, ethical, and health communication but do not yet include topics related to oral healthcare. The inclusion of oral health education into the IPE curriculum can increase awareness and interest in understanding the importance of oral health to systemic health [24,25]. Studies showed that medical practitioners who understand oral healthcare were more likely to provide counsel on oral–systemic health connections [26]. A study in the US reported that the greatest barrier to interprofessional collaboration is the lack of formal relationships [14]. Thus, with oral healthcare IPE being organized early and regularly during university years, students will have chances to build rapport and receive training that they can subsequently apply when they work professionally.

This study found that the NS group showed the least awareness, attitude, and perception of oral healthcare. Moreover, only 45.9% of NS respondents felt that hospitalized patients needed oral healthcare. Bhagat et al. claimed that there was a gap in the nursing curriculum related to oral health, although they also believed that providing oral care is important [27]. From their study, many educators reported that oral health education was not given their full attention, and oral healthcare was an elective course [27]. This could be why first-year NS have no prior knowledge about the importance of oral healthcare.

Nurses hold a very important role, especially in hospital settings for oral healthcare. For example, patients in ICU who are being cared for by nurses on a daily basis may need specific care and demand the highest standards of professional care [8]. A good understanding of oral healthcare is considered vital to care for hospitalized patients, yet it is often neglected [28]. However, six out of eleven studies in a systematic review conducted by Bhagat et al. suggested an interprofessional oral health education model [29], so this could be the solution to improve NS oral healthcare knowledge.

The strength of this study is that there were background variations of healthcare students, i.e., dental, medical, and nursing schools, and the high response rate from each group. Although there have been many published studies related to IPE and reports that Asian students were ready for IPE [30,31], to the best of our knowledge, this study was the first study of oral healthcare IPE in undergraduate students in South East Asia. Oral health is an important part of general health, so collaborative oral healthcare is in the best interest of every health profession. The limitation of this study was that the respondents were first-year students. But, this was also an opportunity since we could investigate their baseline awareness, perception, and attitudes to identify the weak points and give input to the development of an oral healthcare IPE curriculum.

## 5. Conclusions

This study showed that first-year NS had lower awareness, attitudes, and perceptions of oral healthcare compared to DS and MS. However, several aspects of oral healthcare still have room for improvement as well in DS and MS. The result of this study can serve as a suggestion to formulate IPE modules and curricula about oral healthcare that will be given to all students in dental, medical, and nursing schools. It is imperative to improve their awareness, attitudes, and perceptions of oral healthcare as they are going to work in a team to manage patients’ holistic wellbeing.

## Figures and Tables

**Table 1 dentistry-11-00169-t001:** Respondents’ characteristics.

Characteristics	N (N = 442)	%
Age		
<18	72	16.3
≥18	370	83.7
Sex		
Female	339	76.7
Male	103	23.3
Study program		
DS	117	26.5
MS	214	48.4
NS	111	25.1

DS = dental students, MS = medical students, NS = nursing students.

**Table 2 dentistry-11-00169-t002:** Students’ awareness toward oral healthcare.

	Total N (%) (N = 442)	DS N (%) (N = 117)	MS N (%) (N = 214)	NS N (%) (N = 111)	*p*
Knowledge regarding oral health care					0.053
Very much	27 (6.1)	7 (6)	12 (5.6)	8 (7.2)	
Somewhat	198 (44.8)	67 (57.3)	89 (41.6)	42 (37.8)	
A little	178 (40.3)	38 (32.5)	90 (42.1)	50 (45)	
Not very much	39 (8.8)	5 (4.3)	23 (10.7)	11 (9.9)	
Not at all	0 (0)	0 (0)	0 (0)	0 (0)	
Feeling that the effectiveness of oral healthcare is generally known to people					0.152
Very much	19 (4.3)	3 (2.6)	7 (3.3)	9 (8.1)	
Somewhat	41 (9.3)	12 (10.3)	16 (7.5)	13 (11.7)	
A little	140 (31.7)	33 (28.2)	79 (36.9)	28 (25.2)	
Not very much	217 (49.1)	60 (51.3)	103 (48.1)	54 (48.6)	
Not at all	25 (5.7)	9 (7.7)	9 (4.2)	7 (6.3)	
Interest in oral healthcare					<0.001 *
Very much	137 (31)	56 (47.9)	45 (21)	36 (32.4)	
Somewhat	173 (39.1)	48 (41)	77 (36)	48 (43.2)	
A little	99 (22.4)	12 (10.3)	69 (32.2)	18 (16.2)	
Not very much	29 (6.6)	1 (0.9)	20 (9.3)	8 (7.2)	
Not at all	4 (0.9)	0 (0)	3 (1.4)	1 (0.9)	
Interest in practicing oral healthcare after obtaining professional qualifications					<0.001 *
Very much	113 (25.6)	61 (52.1)	26 (12.1)	26 (23.4)	
Somewhat	178 (40.3)	43 (36.8)	86 (40.2)	49 (44.1)	
A little	88 (19.9)	12 (10.3)	53 (24.8)	23 (20.7)	
Not very much	44 (10)	0 (0)	35 (16.4)	9 (8.1)	
Not at all	19 (4.3)	1 (0.9)	14 (6.5)	4 (3.6)	
The proportion of the respondent’s duties that would account for practicing oral healthcare after obtaining their professional qualification					<0.001 *
<25%	118 (26.7)	3 (2.6)	95 (44.4)	20 (18)	
25–49%	107 (24.2)	16 (13.7)	62 (29)	29 (26.1)	
50–74%	138 (31.2)	47 (40.2)	44 (20.6)	47 (42.3)	
≥75%	79 (17.9)	51 (43.6)	13 (6.1)	15 (13.5)	

* Chi-Square test, *p* < 0.05. DS = dental students, MS = medical students, NS = nursing students.

**Table 3 dentistry-11-00169-t003:** Student’s perceptions toward oral healthcare.

	Total N (%) (N = 442)	DS N (%) (N = 117)	MS N (%) (N = 214)	NS N (%) (N = 111)	*p*
Perception regarding the required knowledge to practice oral healthcare					
General dentistry	428 (96.8)	116 (99.1)	207 (96.7)	105 (94.6)	0.136
General medicine	263 (59.5)	67 (57.3)	142 (66.4)	54 (48.6)	0.007 *
Geriatrics	190 (43)	53 (45.3)	73 (34.1)	64 (57.7)	<0.001 *
Nursing	143 (32.4)	54 (46.2)	65 (30.4)	24 (21.6)	<0.001 *
Do not know	10 (2.3)	0 (0)	5 (2.3)	5 (4.5)	0.056
Individuals who should receive oral healthcare					
Older adults who need nursing care	338 (76.5)	93 (79.5)	172 (80.4)	73 (65.8)	0.009 *
Healthy older adults	294 (66.5)	80 (68.4)	153 (71.5)	61 (55)	0.010 *
Patients in hospital wards	253 (57.2)	65 (55.6)	137 (64)	51 (45.9)	0.007 *
Healthy people except for older adult	195 (44.1)	46 (39.3)	104 (48.6)	45 (40.5)	0.182
Cancer patient	205 (46.4)	53 (45.3)	110 (51.4)	42 (37.8)	0.065
Do not know	10 (2.3)	1 (0.9)	4 (1.9)	5 (4.5)	0.211
Places where oral healthcare should be provided					
Long-term care facilities	316 (71.5)	93 (79.5)	154 (72)	69 (62.2)	0.015 *
At home	272 (61.5)	74 (63.2)	134 (62.6)	64 (57.7)	0.62
Pediatric wards	228 (51.6)	59 (50.4)	115 (53.7)	54 (48.6)	0.656
Recovery phase Rehabilitation wards	140 (31.7)	42 (35.9)	72 (33.6)	26 (23.4)	0.089
Cancer hospitals	88 (19.9)	33 (28.2)	42 (19.6)	13 (11.7)	0.008 *
Hospices	213 (48.2)	59 (50.4)	106 (49.5)	48 (43.2)	0.478
Acute Care Hospital (including the ICU)	90 (20.4)	25 (21.4)	45 (21)	20 (18)	0.776
Maternity wards	46 (10.4)	15 (12.8)	25 (11.7)	6 (5.4)	0.13
Psychiatric wards	56 (12.7)	18 (15.4)	30 (14)	8 (7.2)	0.127
Do not know	22 (5)	5 (4.3)	7 (3.3)	10 (9)	0.072
Things that are influenced by oral healthcare					
Prevention of dental caries	419 (94.8)	115 (98.3)	204 (95.3)	100 (90.1)	0.018 *
Prevention of periodontal disease	337 (76.2)	102 (87.2)	176 (82.2)	59 (53.2)	<0.001 *
Prevention of general disease	262 (59.3)	76 (65)	139 (65)	47 (42.3)	<0.001 *
Prevention of cardiovascular disease	90 (20.4)	34 (29.1)	43 (20.1)	13 (11.7)	0.005 *
Prevention of aspiration pneumonia	63 (14.3)	19 (16.2)	34 (15.9)	10 (9)	0.188
Care prevention (prevention of becoming frail)	94 (21.3)	29 (24.8)	47 (22)	18 (16.2)	0.27
Improvement of anorexia	93 (21)	26 (22.2)	53 (24.8)	14 (12.6)	0.036 *
Do not know	13 (2.9)	1 (0.9)	6 (2.8)	6 (5.4)	0.134

* Chi-Square test, *p* < 0.05. DS = dental students, MS = medical students, NS = nursing students.

**Table 4 dentistry-11-00169-t004:** Students’ attitudes regarding learning oral healthcare in lecture or practice.

	Total N (%) (N = 442)	DS N (%) (N = 117)	MS N (%) (N = 214)	NS N (%) (N = 111)	*p*
In lectures					
Tooth brushing	334 (75.6)	96 (82.1)	170 (79.4)	68 (61.3)	<0.001 *
Use of an interspace brush	293 (66.3)	92 (78.6)	142 (66.4)	59 (53.2)	<0.001 *
Swabbing oral soft tissue	280 (63.3)	79 (67.5)	137 (64)	64 (57.7)	0.291
Support of tooth brushing	226 (51.1)	71 (60.7)	111 (51.9)	44 (39.6)	0.006 *
Gargling	276 (62.4)	82 (70.1)	140 (65.4)	54 (48.6)	0.002 *
Removing tongue coating	335 (75.8)	92 (78.6)	162 (75.7)	81 (73)	0.608
Domiciliary dental care	160 (36.2)	48 (41)	73 (34.1)	39 (35.1)	0.441
Cleaning dentures	249 (56.3)	86 (73.5)	113 (52.8)	50 (45)	<0.001 *
Salivary gland massage	182 (41.2)	48 (41)	96 (44.9)	38 (34.2)	0.182
Indirect training in swallowing	137 (31)	39 (33.3)	78 (36.4)	20 (18)	0.002 *
Direct training in swallowing (using foods and drinks)	144 (32.6)	41 (35)	80 (37.4)	23 (20.7)	0.008 *
Oral management in the perioperative ward	239 (54.1)	63 (53.8)	118 (55.1)	58 (52.3)	0.883
Language training	91 (20.6)	38 (32.5)	35 (16.4)	18 (16.2)	0.001 *
Do not know	8 (1.8)	0 (0)	5 (2.3)	3 (2.7)	0.189
In practice					
Tooth brushing	356 (80.5)	98 (83.8)	173 (80.8)	85 (76.6)	0.387
Use of an interspace brush	294 (66.5)	89 (76.1)	146 (68.2)	59 (53.2)	0.001 *
Swabbing oral soft tissue	271 (61.3)	74 (63.2)	133 (62.1)	64 (57.7)	0.646
Support of tooth brushing	239 (54.1)	68 (58.1)	118 (55.1)	53 (47.7)	0.265
Gargling	243 (55)	67 (57.3)	127 (59.3)	49 (44.1)	0.028 *
Removing tongue coating	315 (71.3)	85 (72.6)	159 (74.3)	71 (64)	0.138
Domiciliary dental care	141 (31.9)	39 (33.3)	77 (36)	25 (22.5)	0.044 *
Cleaning dentures	279 (63.1)	81 (69.2)	132 (61.7)	66 (59.5)	0.258
Salivary gland massage	223 (50.5)	62 (53)	111 (51.9)	50 (45)	0.412
Indirect training in swallowing	170 (38.5)	50 (42.7)	95 (44.4)	25 (22.5)	<0.001 *
Direct training in swallowing (using foods and drinks)	174 (39.4)	48 (41)	96 (44.9)	30 (27)	0.007 *
Oral management in the perioperative ward	192 (43.4)	38 (32.5)	114 (53.3)	40 (36)	<0.001 *
Language training	80 (18.1)	22 (18.8)	40 (18.7)	18 (16.2)	0.837
Do not know	7 (1.6)	0 (0)	5 (2.3)	2 (1.8)	0.29

* Chi-Square test, *p* < 0.05. DS = dental students, MS = medical students, NS = nursing students.

## Data Availability

The datasets used and analyzed during this current study are not publicly available due to a subsequent undergoing study but are available from the corresponding author upon reasonable request.

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
