# Peer review of "Awareness, Attitudes, and Perceptions of Oral Healthcare among First Year Dental, Medical, and Nursing Students"

_dentistry, 2023, doi:10.3390/dj11070169_

Round 1

Reviewer 1 Report

This cross-sectional study aimed to investigate the level of awareness, attitudes, and perceptions about oral healthcare among university students of dentistry, medicine, and nursing in Indonesia. My objections are concerned with sample collection. The sample size calculation is missing, also the test of the sample distribution. The authors concluded that first-year nursing students had lower awareness, attitudes, and perceptions of oral healthcare compared to dental and medical students,  so in the Discussion section, it should be explained.

Author Response

Reviewer Comment

This cross-sectional study aimed to investigate the level of awareness, attitudes, and perceptions about oral healthcare among university students of dentistry, medicine, and nursing in Indonesia. My objections are concerned with sample collection. The sample size calculation is missing, also the test of the sample distribution. The authors concluded that first-year nursing students had lower awareness, attitudes, and perceptions of oral healthcare compared to dental and medical students,  so in the Discussion section, it should be explained.

Sample size was obtained using Slovin’s formula has been added in the methods.

The data analysis was mostly comparing between categories using frequency (proportion). For this analysis, sample distribution is not necessary thus not mentioned in the manuscript.

Nursing students had the least understanding of oral health care compared to medical and dental students. The discussion had been rewritten to show that nursing student curriculum has been reported to be inadequate by previous reports which could be the reason why such result was found in this study. Thus, supported by previous studies as well, we suggest that oral health care should be introduced and taught in interprofessional education in students of nursing, medical, and dental schools.

Reviewer 2 Report

In general,It is a methodologically well-executed investigation. However, it is recommended to review the grammar and, if possible, add how the sample size was obtained and detail the process of adaptation of the instrument.

Author Response

Reviewer Comment

In general,It is a methodologically well-executed investigation. However, it is recommended to review the grammar and, if possible, add how the sample size was obtained and detail the process of adaptation of the instrument.

English grammar has been reviewed using professional English editor and has been reviewed once again by authors.

Sample size was obtained using Slovin’s formula has been added in the methods.

Brief details of cross-cultural adaptation of the instrument has been added in the methods.

Round 2

Reviewer 1 Report

The authors have accept my suggestions.